# City-Level Travel Time and Individual Dietary Consumption in Latin American Cities: Results from the SALURBAL Study

**DOI:** 10.3390/ijerph192013443

**Published:** 2022-10-18

**Authors:** Joanna M. N. Guimarães, Binod Acharya, Kari Moore, Nancy López-Olmedo, Mariana Carvalho de Menezes, Dalia Stern, Amélia Augusta de Lima Friche, Xize Wang, Xavier Delclòs-Alió, Daniel A. Rodriguez, Olga Lucia Sarmiento, Leticia de Oliveira Cardoso

**Affiliations:** 1Epidemiology Department, National School of Public Health, Oswaldo Cruz Foundation, Rio de Janeiro 21041-210, Brazil; 2Urban Health Collaborative, Dornsife School of Public Health, Drexel University, Philadelphia, PA 19104, USA; 3Population Health Research Center, National Instituto Nacional de Salud Pública, Cuernavaca 62100, Mexico; 4Department of Clinical and Social Nutrition, Federal University of Ouro Preto, Ouro Preto 35400-000, Brazil; 5CONACyT-Population Health Research Center, National Instituto Nacional de Salud Pública, Cuernavaca 62100, Mexico; 6Department of Speech, Language and Audiology Sciences, Observatory for Urban Health in Belo Horizonte, School of Medicine, Federal University of Minas Gerais, Belo Horizonte 30310-692, Brazil; 7Department of Real Estate, National University of Singapore, Singapore 119245, Singapore; 8Research Group on Territorial Analysis and Tourism Studies (GRATET), Department of Geography, Universitat Rovira i Virgili, 43480 Vila-seca, Catalonia, Spain; 9Institute of Transportation Studies, Department of City and Regional Planning, University of California, Berkeley, CA 94720, USA; 10Department of Public Health, School of Medicine, Universidad de Los Andes, Bogota 111711, Colombia

**Keywords:** urban health, mobility systems, city level travel time, diet

## Abstract

There is limited empirical evidence on how travel time affects dietary patterns, and even less in Latin American cities (LACs). Using data from 181 LACs, we investigated whether longer travel times at the city level are associated with lower consumption of vegetables and higher consumption of sugar-sweetened beverages and if this association differs by city size. Travel time was measured as the average city-level travel time during peak hours and city-level travel delay time was measured as the average increase in travel time due to congestion on the street network during peak hours. Vegetables and sugar-sweetened beverages consumption were classified according to the frequency of consumption in days/week (5–7: “frequent”, 2–4: “medium”, and ≤1: “rare”). We estimate multilevel ordinal logistic regression modeling for pooled samples and stratified by city size. Higher travel time (Odds Ratio (OR) = 0.65; 95% Confidence Interval (CI) 0.49–0.87) and delay time (OR = 0.57; CI 0.34–0.97) were associated with lower odds of frequent vegetable consumption. For a rare SSB consumption, we observed an inverse association with the delay time (OR = 0.65; CI 0.44–0.97). Analysis stratified by city size show that these associations were significant only in larger cities. Our results suggest that travel time and travel delay can be potential urban determinants of food consumption.

## 1. Introduction

Unhealthy diets are a risk factor for type II diabetes, cardiovascular diseases, and other non-communicable chronic diseases [1,2,3]. A growing body of evidence from high-income countries suggests that built environments, such as those with poor walkability and low access to supermarkets, play a role in shaping unhealthy dietary behaviors such as low consumption of fruit and vegetables and high consumption of sugar-sweetened beverages (SSBs) [4,5,6]. In cities, the built environment encompasses the characteristics of human and natural environments such as how buildings and spaces are designed, the land uses in them, and transportation options available [7]. Together, these systems determine the location of activities in space and the transportation means to access them.

Travel time is a measure that often summarizes the ease of getting to destinations for a particular mode of transportation. Long travel times have been observed due to increased urbanization [8] because the location of certain activities is further away in larger cities. Additionally, one of the adverse effects of increasing motor vehicle use is its contribution to congestion and, consequently, to longer travel times [9]. Long travel durations may further affect other activities in daily life, such as healthy behaviors. Long travel times have been associated with poor dietary patterns in the US population [10,11]. A potential explanation for this association is that more time spent traveling can translate into less time for buying food and cooking at home, leading to meals involving low preparation time (e.g., pre-prepared foods) and greater consumption of ultra-processed foods [11,12]. For instance, the American Time Use Survey (2003–2010) showed that spending an additional 120 min daily commuting above average was associated with a 5.6% decrease in food preparation time [11]. In addition, a recent study by Langellier et al. (2019) identified a causal link between travel time and consumption of ultra-processed foods.

Despite the importance of studying whether travel time may affect dietary behaviors, the empirical evidence is limited, and even less is known about it in Latin American cities (LACs). A previous study conducted in 178 LACs explored the relationship between city-level travel time by car and individual obesity and found no statistically significant association [13]. The authors explained that this association maybe more complex in LACs because the use of means of transportation shows a different pattern in comparison with cities in Europe and the US. For instance, a greater proportion of Latin Americans commute from their residence to their workplace by public transport or by foot, while fewer people use private transport than in high-income contexts [14]. However, in this work the authors did not analyze the association between city level travel time and diet or physical activity (risk factors for diabetes or obesity). Moreover, trends in dietary intake in Latin American countries suggest a shift toward unhealthy dietary patterns [1,15,16]. As a consequence, daily consumption of fruit and vegetables in all Latin American countries is estimated to be under the recommended 400 grammes per person, with an average per capita consumption of fruits and vegetables being 120.6 and 104.0 grammes, respectively. In contrast, the average consumption of SSB is almost 500 grammes per person per day, which makes it very likely that most people in the region consume over 50 grammes of the maximum recommended amount of sugar per day [17]. The Latin American region is one of the world’s most urbanized and unequal areas, with large heterogeneity in built environments [18]. Furthermore, there is a strong increase in ownership and use of cars and motorcycles in the region, especially in larger cities [9].

These features make Latin American cities uniquely positioned to investigate how travel time may be associated with dietary habits. Using secondary data from different sources and a multilevel approach, this study investigated whether longer travel times at the city level are associated with lower consumption of vegetables (as proxy of healthy diet) and higher consumption of SSB (as proxy of unhealthy diet) among individuals and if this association differs by city size.

## 2. Materials and Methods

### 2.1. Study Population

This cross-sectional study used harmonized data for adults 18 years or older from national health surveys in 181 Latin American cities from Brazil (Pesquisa Nacional de Saúde (PNS) 2013), Chile (Encuesta Nacional de Salud (ENS) 2010), Colombia (Encuesta Nacional de la Situation Nutricional en Colombia (ENSIN) 2010), Mexico (Encuesta Nacional de Salud y Nutrición (ENSANUT) 2012), and Peru (Encuesta Nacional de Demografia y Salud (ENDES) 2016). Cities were defined as groups of administrative units that encompass the urban extent as determined by satellite imagery [19]. Of the 60,933 survey participants, 57,170 and 42,117 had complete information on all individual and city-level variables and were included for vegetable consumption and SSB consumption analyses, respectively.

### 2.2. Measures

Travel time variables were the city-level average (1) travel time and (2) delay time. Average travel time by car during the peak hour was measured in minutes. Travel times were calculated in 2018 using the Google Maps Distance Matrix API [20] by selecting 30 random origin–destination pairs in each city, considering only the urban extent instead of the entire administrative area. Four morning peak-hour measurements (6:30, 7:30, 8:30, and 9:30 a.m.) and three afternoon peak-hour measurements (5:30, 6:30, 7:30 p.m.) of a typical weekday (Tuesday–Thursday) were taken. Travel delay time in minutes was obtained by subtracting the average travel time during the peak hour from free flow travel time, measured around 2 and 3 a.m. (travel time without traffic).

Vegetable and SSB individual consumption (outcomes) were assessed through a food frequency questionnaire (FFQ) (Mexico) or dietary survey questions (other countries) (Appendix A). The number of days per week with consumption of vegetables (available for all surveys) or SSB (available for Brazil, Colombia, and Mexico) was categorized as: rare (≤1), medium (2–4), or frequent (5–7 days/week). This ranked variable was used as a proxy for a healthy diet (the higher the frequency of vegetable consumption, the healthier the diet; the opposite was true for SSB consumption). We chose to assess the consumption of vegetables rather than fruits because vegetables are usually consumed as part of meals and require culinary preparation in line with our research question.

Individual-level covariates were self-reported sex, age (continuous), education (less than primary, completed primary, completed secondary, and completed university) and household car ownership (yes/no). City-level covariates were the size of urban built-up area (in hectares), population density (persons per hectare, obtained by dividing the city population by the total built-up area in the city), intersection density (streets intersections per km^2^, in which higher values indicate more connected streets), adjusted gasoline price (% of the country monthly minimum wage needed to purchase 10 gallons of gasoline, in USD), presence of mass public transportation (Bus Rapid Transit or subway, yes/no) and a social environment index [21], a combination of Z-scores of city features including the % of the population 25 years of age or older who completed primary education or above, % of households with access to piped water, % of households with access to a municipal sewage network, and % of households with more than three people per room (overcrowding).

### 2.3. Statistical Analysis

Covariates were compared across categories of vegetable and SSB consumption. Multilevel ordinal logistic regression models (individuals nested within cities) with a random intercept for each city were used, and odds ratios (ORs) and 95% confidence intervals (CI) were estimated. Exposures (average travel time and delay time) and outcomes (frequent vegetable consumption and rare SSB consumption) variables were analyzed in separate models. We first estimated associations between city-level travel time and individual dietary consumption adjusted for individual-level confounders: age, sex, education, and car ownership (Model 1). We then added city-level confounders: population density, intersection density, adjusted gas price, presence of mass transportation and city size (Model 2), and the social environment index (Model 3). Exposure variables were re-scaled to 10 min units to facilitate interpretation. We hypothesized that the association between travel time variables and outcomes might be different depending on the city size as it can be considered a proxy of other city characteristics that affect the association between travel time and the outcomes (for example number of cars in city level, mobility systems regulations as speed limits, car license plate rotation, and so on). Thus, we categorized city size into three groups (<10,000/10,000–50,000/>50,000 hectares (ha)) to obtain a comparable number of respondents in each group and performed stratified analyses by this variable. Analyses were performed using SAS 9.4 software (SAS Inc., Cary, NC, USA).

## 3. Results

The vegetable consumption analytic sample was comprised of 57,170 respondents from national health surveys from Brazil, Colombia, Mexico, Chile, and Peru. Respondents were distributed over 181 cities, with a median of 52 respondents per city (inter-quartile range = 269). The prevalence of rare (≤1 day/week), medium (2–4), and frequent (5–7) consumption of vegetables was 18%, 36%, and 46%, respectively. In general, female, older, and more educated individuals were more likely to report a greater vegetable consumption. Regarding the city-level characteristics, travel time in minutes was higher for those respondents with more frequent consumption of vegetables (mean = 30.0, SD = 13.4) than for those with rare consumption (mean = 26.0, SD = 15.4). In contrast, travel delay time was lower for those with frequent (mean = 5.5, SD = 4.2) than for those with rare (mean = 6.1, SD = 6.7) vegetable consumption. Respondents with higher vegetable consumption were more likely to live in larger cities, with higher intersection density, higher presence of mass transportation, and better socioeconomic conditions (Table 1).

For the SSB analysis, the sample comprised of 42,117 respondents from health surveys from Brazil, Colombia, and Mexico, distributed in 137 cities (median of 31 respondents per city, inter-quartile range = 113). The prevalence of frequent (5–7 day/week), medium (2–4), and rare (≤1) consumption of SSB was 22.9%, 27.9%, and 49.2%, respectively. Female, older, and more educated individuals were more likely to have a lower SSB consumption, although lower-educated individuals also had low SSB consumption. Travel time in minutes was higher for those respondents with rare consumption of SSB (mean = 30.7, SD = 11.9) than for those with more frequent SSB consumption. For travel delay time, the pattern was less marked, with respondents with a medium consumption of SSB (mean = 5.0, SD = 2.8) showing slightly lower delay time compared with those with frequent (mean = 5.2, SD = 3.0) or rare (mean = 5.2, SD = 3.0) SSB consumption. In addition, respondents with rare consumption of SSB were more likely to live in larger cities, with higher population and intersection density, with a higher presence of mass transportation and in better socioeconomic conditions (Table 2).

After adjusting for the individual confounders age, sex, education, car ownership, and city-level confounders population density, intersection density, adjusted gas price, mass transportation presence, and city size, higher levels of both travel time and delay time were associated with lower odds of frequent vegetable consumption (OR = 0.65 95% CI 0.49–0.87 and OR = 0.57 95% CI 0.34–0.97, respectively) (Table 3, Model 2). However, travel time and delay time were not associated with rare consumption of SSB (OR = 1.01 95% CI 0.81–1.27 and OR = 0.67 95% CI 0.45–1.00, respectively). Further adjustment for the city-level social environment index (Model 3) attenuated the association between travel time and frequent vegetable consumption (OR = 0.89 95% CI 0.68–1.16), but the association between delay time and frequent vegetable consumption was such that for a 10 min higher delay time, there was 45% lower odds of having a frequent versus rare consumption of vegetables (OR = 0.55 95% CI 0.33–0.91). Association between delay time and SSB indicated that, for a 10 min higher delay time, there was 35% lower odds of having rare versus frequent consumption of SSB (OR = 0.65 95% CI 0.44–0.97) (Table 3, Model 3).

Although the interaction term between the exposures and the city was not significant in the models, the analyses stratified by city size area tertiles showed that longer travel time was associated with lower odds of frequent vegetable consumption, but only in the largest cities’ tertile (OR = 0.60 95% CI 0.41–0.89) (Figure 1, Panel A, M3). Travel time was not associated with rare SSB consumption, in any of the city size tertiles (Figure 1, Panel B, M3). Longer delay time was associated with 69% lower odds (OR = 0.31 95% CI 0.21–0.46) and 78% lower odds (OR = 0.22 95% CI 0.11–0.45) of having a more frequent consumption of vegetables (Figure 2, Panel A, M3) and a rarer consumption of SSB (Figure 2, Panel B, M3), respectively, but only at the largest cities’ tertile.

## 4. Discussion

To our knowledge, this is the first study to examine associations between city level travel time and individual diet quality in LAC using a large and geographically comprehensive sample. Our results partially support our hypothesis that longer average travel times at the city level are inversely associated with a healthy dietary pattern (i.e., a more frequent consumption of vegetables and a rare consumption of SSB after controlling by confounders). We also demonstrated that the observed associations are clearer when the exposure variable analyzed is the delay time, which directly measures how long the traffic increases the travel time and are stronger in larger cities.

These findings are consistent with the view that cities are complex environments with many variables interacting in feedback dynamics [12]. For example, additional time spent in traffic can reduce the time available for other activities, such as cooking [11,12]. Indeed, time availability is a key determinant of home cooking [22]. A previous study showed that a greater amount of time spent on home food preparation is associated with more frequent consumption of healthy food markers that requires preparation, such as vegetables, salads, and prepared fruit juices [23]. With decreased time for health-related activities such as home cooking [11], a potential maleficent feedback loop is a greater consumption of ultra-processed foods as they are mostly highly convenient ready-to-consume food items [12]. Although we were not able to demonstrate a relationship between the travel time and the consumption of an ultra-processed food marker, we observed a significant association with the delay time after adjustment for the city-level social environment index. A possible explanation for the less evident relationship with the unhealthy diet is that we used SSB consumption to proxy for unhealthy diet. It was the best available and comparable marker of ultra-processed food consumption across the countries assessed, but it probably is not the most discriminative variable in this research context. After all, beverages do not directly compete with homemade food in terms of required preparation time as they can be consumed anywhere.

Another possible reason for us to find an association of SSB consumption only with delay time is that this exposure variable corresponds to the difference between the average travel time during the peak hour and free-flow travel time. Therefore, it expresses how much travel time is lost due to traffic congestion and is possibly able to more directly capture people’s difficulty to plan ahead to cook and consume fresh food. Although some congestion is recurrent, up to 60% of all travel delay is non-recurrent and thus hard to anticipate [24], which may explain our findings. Together with studies from high-income countries, which found a link between travel time variables and indicators of metabolic risk such as BMI, waist circumference, blood pressure, and fasting plasma glucose [25,26], the present investigation shows that cities’ mobility systems may be a relevant entry point for policymakers when designing and implementing health policies and programs. Nevertheless, a recent study using data from 178 LAC found no association between city-level travel time and individual obesity or diabetes [13]. The contrast between their findings and our results might be due to sample differences (e.g., their study participants were younger and travel time in minutes was lower than ours) and differences in exposures variables; they did not include travel delay time that seems to be more specific to capture the association of interest. Another point to consider, the referred diabetes diagnosis, can be more susceptible to information bias compared with diet heathy indicators, which can underestimate the association measure. Moreover, our outcome, dietary intake, represents a more proximal factor (or an intermediate factor) in the complex causal chain than chronic diseases such as obesity and diabetes and because of this are more likely to be identified.

We also found that the associations observed were strengthened in the largest cities, even for a rarer consumption of SSB. Latin America is one of the most urbanized regions globally [27], its urban spatial development is characterized by a high concentration of built-up area in the cities’ core. Moreover, there has been a great increase in passenger vehicle motorization rates, which is worrying considering its cities’ high street density [28]. Thus, city size can be considered a proxy of features that may affect travel time such as urban traffic, proportions of circulating vehicles, and urban street density. Considering the particularly high rates of obesity and chronic diseases related to diet in Latin America, these findings are worrisome because many of the world’s megacities as well as several small and middle-sized cities in rapid growth are located in the region [18,27].

Finally, we were able to identify the reported associations even though the travel and delay time calculations only took into account the time estimates for motor vehicles (e.g., cars, buses) in mixed traffic. Public transportation and private vehicles continue to be some of the two most popular means of motor vehicle travel in Latin America, with vehicle motorization rates continually increasing along with urbanization [28]. Walking and cycling are also important modes of transportation in the region, representing 10.6% and 6.3% of the total travel time [29], and might be especially relevant for smaller Latin American cities. However, our metrics did not capture the influence of active modes of transportation on city-level travel time.

Despite our relevant findings, this study has some important limitations to consider. First, there is temporal misalignment between the exposure and outcome variables. The city-level average travel time and delay time were calculated in 2018 using the Google Maps Distance Matrix API, while data on vegetable and SSB consumption, as well as the individual level covariates, come from national surveys whose date of collection ranged from 2010 to 2016 among the countries assessed. However, the general trend in Latin American countries is towards a decrease in vegetable consumption and an increase in SSB, as well as an increase in travel time, so the actual magnitude and direction of the association is likely similar to what we are observing. Second, we hypothesized that there is an association between travel time and diet, but we were only able to capture a city-level average of travel time during the peak hour in each city. Considering this is an ecological exposure, we are unable to make inferences at the individual level. Third, even though we adjusted the analysis by individual car ownership, the city-level estimation of travel time in traffic only considers motorized travel in mixed traffic. Estimates are higher for those using public transportation due to passenger drop off and pick up, and are incorrect for active transportation users, as well as users of transportation services reliant on exclusive rights of way (e.g., subway). Finally, regarding the outcome, the best diet variables available and comparable among countries’ health surveys were proxies for healthy and unhealthy diets. Thus, we only analyzed a few food markers that were measured as the weekly frequency of consumption, not the amount consumed. Although vegetable intake is well-established as a proxy for a healthy diet, unhealthy food markers other than SSB such as ready-to-eat food, savory snacks, or candies consumption can help to better discriminate groups with different levels of ultra-processed foods intake. Future research that aims to explore this association in the individual level should investigate this hypothesis using longitudinal data (cohort studies) guaranteeing the temporal preceding of exposure and diet variables. Moreover, it should include individual information of travel or commute time and type of transportation, and a more broad diet questionnaire. For future ecological or multilevel studies, the guarantee of temporality between exposures and outcomes is a key point.

Our main strength is the novelty of this research. In high-income countries, studies on the relationship between travel time by car and health outcomes have been mostly restricted to one city [30] or state [25,31], and few studies have covered low- and middle-income contexts [32,33]. The ones conducted in Latin America have only assessed specific countries and did not focus directly on the relationship of health outcomes with travel time, but only motor vehicle ownership [34], active commuting [35], or commuting mode [32]. Furthermore, we measured the delay time, which is especially novel as most studies have only assessed the travel time and found very consistent results. Thus, this is the first study on the association between travel time and diet in the region, and probably the most geographically comprehensive to date as it assessed this relationship across countries.

## 5. Conclusions

We observed an association pattern indicating that in cities where the average travel time and delay time are higher, residents have lower odds of having more frequent vegetable consumption and where the average delay time is higher, they have lower odds of having rarer consumption of SSB. Furthermore, we showed that these relationships are stronger in larger cities, where there are more cars circulating and people’s activities tend to be farther away from each other. We draw the attention of policymakers to the importance of urban mobility systems in the quality of the diet, especially in the Latin American context, where the rates of urbanization and passenger vehicle motorization and the prevalence of chronic non-communicable diseases are increasing. Promoting denser, more compact, and mixed-use urban development can allow for shorter daily trips. In turn, the reduction in time spent traveling daily can provide more time available both to access higher quality foods and to prepare healthier meals.

## Figures and Tables

**Figure 1 ijerph-19-13443-f001:**
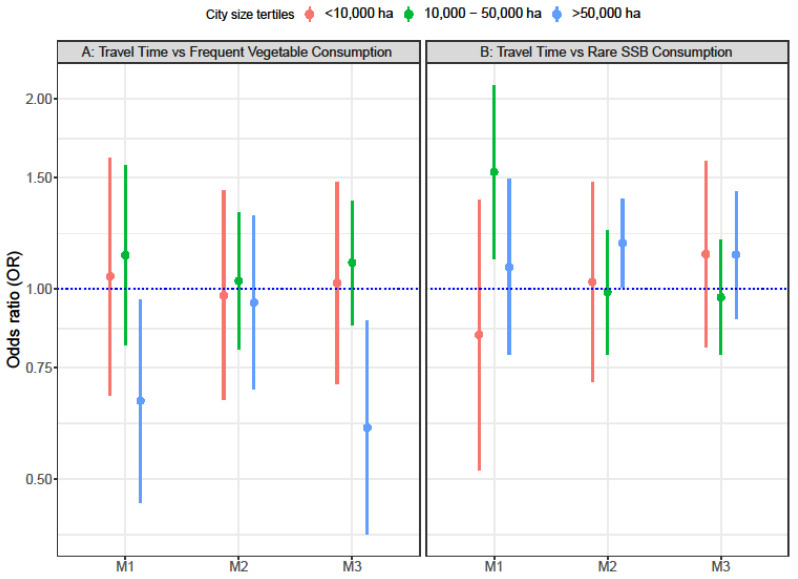
Odds ratios of frequent vegetable consumption (N = 57,170) (**A**) and rare SSB consumption (N = 42,117) (**B**), associated with city level travel time in LA cities, stratified by city size area tertiles. Panel A includes National Health Surveys from Brazil, Colombia, Mexico, Chile, and Peru. Panel B includes National Health Surveys from Brazil, Colombia, and Mexico. M1: + age + sex + education + car ownership. M2: M1 + population density + intersection density + adjusted gas price + presence of mass transportation. M3: M2 + Social environment index.

**Figure 2 ijerph-19-13443-f002:**
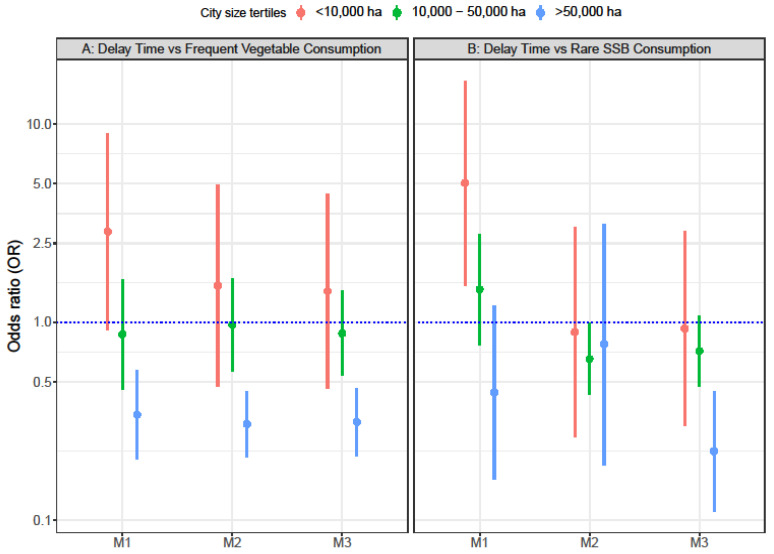
Odds ratios of frequent vegetable frequent consumption (N = 57,170) (**A**) and rare SSB consumption (N = 42,117) (**B**), associated with city level delay time in LA cities, stratified by city size area tertiles. Panel A includes National Health Surveys from Brazil, Colombia, Mexico, Chile, and Peru. Panel B includes National Health Surveys from Brazil, Colombia, and Mexico. M1: + age + sex + education + car ownership. M2: M1 + population density + intersection density + adjusted gas price + presence of mass transportation. M3: M2 + Social environment index.

**Table 1 ijerph-19-13443-t001:** Characteristics of the sample ^a^, overall and by vegetable consumption groups.

Variables	All	Vegetable Consumption (In Days per Week)		
		Rare (≤1)	Medium (2–4)	Frequent (5–7)
N of participants (%)	57,170	10,309 (18.0)	20,606 (36.0)	26,255 (46.0)
Individual level characteristics				
Sex, Female (%)	33,488 (58.6)	5541 (16.5)	11,680 (34.9)	16,267 (48.6)
Age in years, mean (SD)	42.0 (16.4)	39.2 (16.5)	40.9 (16.0)	44.0 (16.5)
Education, %				
University	7665 (13.4)	704 (9.2)	2354 (30.7)	4607 (60.1)
Secondary	24,022 (42.0)	4424 (18.4)	8860 (36.9)	10,738 (44.7)
Primary	13,721 (24.0)	2741 (20.0)	5090 (37.1)	5890 (42.9)
Less than Primary	11,736 (20.5)	2435 (20.7)	4292 (36.6)	5009 (42.7)
Household car ownership,				
Yes (%)	20,190 (35.3)	1933 (9,6)	6423 (31.8)	11,834 (58.6)
City level characteristics				
Travel time in minutes, mean (SD)	28.8 (14.3)	26.0 (15.4)	28.8 (14.6)	30.0 (13.4)
Travel delay time in minutes, mean (SD)	5.9 (5.3)	6.1 (6.7)	6.1 (5.7)	5.5 (4.2)
City size, mean (SD)	40,004 (47,072)	25,027 (34,116)	37,525 (45,750)	47,830 (50,705)
Population density, mean (SD)	9625 (3990)	11,013 (4274)	9943 (4104)	8831 (3578)
Intersection density, mean (SD)	14.5 (11.7)	12.6 (12.0)	14.4 (11.7)	15.4 (11.4)
Adjusted gas price, mean (SD)	0.04 (0.01)	0.04 (0.01)	0.04 (0.01)	0.03 (0.01)
Presence of mass transport, Yes (%)	32,555 (56.9)	4289 (13.2)	11,232 (34.5)	17,034 (52.3)
Social environment index ^b^, mean (SD)	0.11 (0.53)	−0.02 (0.54)	0.06 (0.55)	0.19 (0.50)

^a^ Includes all countries Brazil (2013), Colombia (2010), Mexico (2012), Chile (2010), and Peru (2016). ^b^ Higher score indicates better social environment.

**Table 2 ijerph-19-13443-t002:** Characteristics of the sample ^a^, overall and by sugar-sweetened beverages (SSBs) consumption groups.

Variables	All	SSB Consumption (In Days per Week)		
		Frequent (5–7)	Medium (2–4)	Rare (≤1)
Number of participants (%)	42,117	9626 (22.9)	11,752 (27.9)	20,739 (49.2)
Individual level characteristics				
Sex, Female (%)	24,761 (58.8)	4974 (20.1)	6350 (25.6)	13,437 (54.3)
Age in years, mean (SD)	42.3 (16.4)	36.6 (14.2)	39.0 (15.0)	46.9 (16.9)
Education, %				
University	5937 (14.1)	913 (15.4)	1520 (25.6)	3504 (59.0)
Secondary	15,735 (37.4)	3870 (24.6)	4756 (30.2)	7109 (45.2)
Primary	10,574 (25.1)	3031 (28.7)	3042 (28.8)	4501 (42.3)
Less than Primary	9848 (23.4)	1805 (18.3)	2431 (24.7)	5612 (57.0)
Household car ownership, Yes (%)	17,247 (40.9)	3664 (21.2)	4730 (27.4)	8853 (51.3)
City level characteristics				
Travel time in minutes, mean (SD)	30.1 (11.9)	29.8 (12.1)	29.5 (11.8)	30,7 (11.9)
Travel delay time in minutes, mean (SD)	5.1 (3.0)	5.2 (3.0)	5.0 (2.8)	5,2 (3.0)
City size, mean (SD)	47,516 (50,721)	47,607 (50,976)	44,725 (48,325)	49,055 (51,850)
Population density, mean (SD)	8748 (3648)	8447 (3661)	8640 (3568)	8949 (3675)
Intersection density, mean (SD)	14.5 (9.9)	13.7 (10.2)	13,9 (9.8)	15.2 (9.8)
Adjusted gas price, mean (SD)	0.04 (0.01)	0.04 (0.01)	0.04 (0.01)	0.04 (0.01)
Presence of mass transport, Yes (%)	28,189 (66.9)	15,453 (54.8)	9582 (34.0)	3154 (11.2)
Social environment index ^b^, mean (SD)	0.03 (0.52)	0.02 (0.55)	0.0 (0.54)	0.06 (0.50)

^a^ Includes Brazil (2013), Colombia (2010) and Mexico (2012). ^b^ Higher score indicates a better social environment.

**Table 3 ijerph-19-13443-t003:** Odds ratios of individual vegetable frequent consumption (N = 57,170) ^a^ and sugar-sweetened beverage (SSB) rare consumption (N = 42,117) ^b^, associated with city level travel and delay time in LA cities.

Variables	Model 1 ^d^	Model 2 ^e^	Model 3 ^f^
OR for frequent vegetable consumption			
City-level travel time ^c^	1.16 (1.00; 1.34)	0.65 (0.49; 0.87)	0.89 (0.68; 1.16)
City-level delay time ^c^	1.29 (0.82; 2.04)	0.57 (0.34; 0.97)	0.55 (0.33; 0.91)
OR for rare SSB consumption			
City-level travel time ^c^	1.00 (0.88; 1.14)	1.01 (0.81; 1.27)	0.97 (0.77; 1.21)
City-level delay time ^c^	1.26 (0.79; 1.99)	0.67 (0.45; 1.00)	0.65 (0.44; 0.97)

^a^ Includes national health surveys from Brazil, Colombia, Mexico, Chile, and Peru. ^b^ Includes national health surveys from Brazil, Colombia, and Mexico. ^c^ per 10 min increase. ^d^ Model 1: + age + sex + education + car ownership. ^e^ Model 2: M1 + population density + intersection density + adjusted gas price + presence of mass transportation + city size. ^f^ Model 3: M2 + Social environment index.

## Data Availability

SALURBAL acknowledges the contributions of many different agencies in generating, processing, facilitating access to data, or assisting with other aspects of the project. Please visit https://drexel.edu/lac/data-evidence (accessed on 11 October 2022) for a complete list of data sources. The findings of this study and their interpretation are the responsibility of the authors and do not represent the views or interpretations of the institutions or groups that compiled, collected, or provided the data. The use of data from these institutions does not claim or imply that they have participated in, approved, endorsed, or otherwise supported the development of this publication. They are not liable for any errors, omissions, or other defects, or for any actions taken in reliance thereon.

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
