# Peer review of "City-Level Travel Time and Individual Dietary Consumption in Latin American Cities: Results from the SALURBAL Study"

_ijerph, 2022, doi:10.3390/ijerph192013443_

Round 1

Reviewer 1 Report

Comments:

In this work, using data from 197 LAC, the authors investigated whether longer travel times at the city level are associated with lower consumption of vegetables and higher consumption of sugar-sweetened beverages and if this association differs by city size. They observed an association pattern indicating that in cities where the average travel time and delay time are higher, residents have lower odds of having more frequent vegetable consumption and where the average delay time is higher, they have lower odds of having rarer consumption of SSB.

1.     As far as I know, in 2022, there is a new article titled “Is city-level travel time by car associated with individual obesity or diabetes in Latin American cities? Evidence from 178 cities in the SALURBAL project”, asking you to elaborate on the difference between your research and his. His article studies the relationship between ‘city-level travel time by car’ and ‘individual obesity or diabetes’, your article studies the relationship between ‘city-level travel time’ and ‘individual dietary consumption’, what else is there?

2.     If the author can add a literature review in the Section 2, it can better reflect the clearer context of the article and make readers more aware of the current research status of this topic.

3.     Can the author provide the highlights of the article so that readers can get the results of the article more clearly.

4.     As you say in page 9, “a recent study using data from Latin American countries found no association between city-level travel time and individual obesity or diabetes”, the research results of this article and the above results can be compared to illustrate the problem. Please explain more rationales so that readers can fully believe in the research results you have obtained.

5.     In Page 10, You proposed several limitations of this study, but I would prefer that you will be able to point out the limitations and explain the future solutions to these limitations.

Author Response

Rev 1 - Comments:

In this work, using data from 197 LAC, the authors investigated whether longer travel times at the city level are associated with lower consumption of vegetables and higher consumption of sugar-sweetened beverages and if this association differs by city size. They observed an association pattern indicating that in cities where the average travel time and delay time are higher, residents have lower odds of having more frequent vegetable consumption and where the average delay time is higher, they have lower odds of having rarer consumption of SSB.

  1.     As far as I know, in 2022, there is a new article titled “Is city-level travel time by car associated with individual obesity or diabetes in Latin American cities? Evidence from 178 cities in the SALURBAL project”, asking you to elaborate on the difference between your research and his. His article studies the relationship between ‘city-level travel time by car’ and ‘individual obesity or diabetes’, your article studies the relationship between ‘city-level travel time’ and ‘individual dietary consumption’, what else is there?

R: Thank you for your comment. Yes, this paper was recently published and has a different aim. In the previous one we were interested in analyzing the association between ‘city-level travel time by car’ and ‘individual obesity or diabetes’, but we did not observe any association with these outcomes. So, we decided to explore a previous point on this complex web causation of obesity and diabetes, the food consumption. Also, because in a previous qualitative workshop in some Latin American cities the causal loop diagrams(Langellier et al (2019)) had suggested to us that there could exist a correlation between travel time (or commute) and diet quality. We included this study in the ‘Introduction’ (66-69) and expanded this rationality in the ‘Discussion’ section (line 309-316). 

  1.     If the author can add a literature review in the Section 2, it can better reflect the clearer context of the article and make readers more aware of the current research status of this topic.

Thank you for your comment. We follow the sections proposed to the paper format [Introduction, Materials and Methods, Results, Discussion, Conclusions], because of this we did not include a literature review topic. So, we included all the studies we had found in the main library database (Pubmed, Scopus and Web of Science) about the relationship between travel or commute time and diet (or food preparation) in the Introduction section. We also renewed the research strategies and still did not find any new study about this relationship, so we keep the references we cited before.

  1.     Can the author provide the highlights of the article so that readers can get the results of the article more clearly.

Thank you for your suggestion. We did not find a specific topic for highlights in the current format. We provide here the topics we consider the main highlights:

  • This is the first study to examine associations between city-level travel time and individual diet quality in Latin American cities (LAC);
  • We analyze data from 181 LAC - a large and comprehensive sample;
  • Longer average travel times at the city level are inversely associated with a healthy dietary pattern (more frequent vegetable consumption and rare SSB consumption);
  • These associations are stronger in larger cities.

  1.     As you say in page 9, “a recent study using data from Latin American countries found no association between city-level travel time and individual obesity or diabetes”, the research results of this article and the above results can be compared to illustrate the problem. Please explain more rationales so that readers can fully believe in the research results you have obtained. 

Thank you for your suggestion. We added more information and explanations regarding the divergence of our results compared with the previous study mentioned in your comment. Please see line (306-316)

“Nevertheless, a recent study using data from 178 LAC found no association between city-level travel time and individual obesity or diabetes [13]. The contrast between their findings and our results might be due to sample differences (e.g., their study participants were younger and travel time in minutes was lower than ours) and differences in exposures variables, they did not include travel delay time that seems to be more specific to capture the association of interest. Another point to consider, the referred diabetes diagnosis can be more susceptible to information bias compared to diet heathy indicators, which can underestimate the association measure. Moreover, our outcome, dietary intake, represents a more proximal factor (or an intermediate factor) in the complex causal chain than chronic diseases such as obesity and diabetes and because of this are more likely to be identified”.

  1.     In Page 10, You proposed several limitations of this study, but I would prefer that you will be able to point out the limitations and explain the future solutions to these limitations.

Thank you for your suggestion. We added some future solutions that could be followed by future studies (line 358-364).

“Future research that aims to explore this association in the individual level should investigate this hypothesis using longitudinal data (cohort studies) guaranteeing the temporal preceding of exposure and diet variables. Moreover, it should include individual information of travel or commute time and type of transportation and a more broad diet questionnaire information. For future ecological or multilevel studies the guarantee of temporality between exposures and outcomes is a key point."

Reviewer 2 Report

In general, this work is really interesting and well written. My comments aim to increase the scientific soundness and clarity of it.

Line 7 -14  – Please provide more detailed affiliation (Department, Chair etc.).

Line 22, 27 – Please expand SSB, OR and CI

Line 30 – I dont see purpose of using numbers in keywords section.

Line 57, 180, 250 – please use LAC abbreviation as implemented in abstract

Line 180 – please verify whether data were obtained from 181 LAC or 197 LAC (line 17)?

Line 73 – a brief description of methodology applied should follow the aim of the study.

Line 113 – “km2” instead of “km2”

Line 113 – “gas” or “gasoline” ?

Line 138 – please provide details of SAS 9.4 (manufacturer, country of origin, license etc. )

Author Response

Reviewer 2  

In general, this work is really interesting and well written. My comments aim to increase the scientific soundness and clarity of it.

R: We thank four your suggestions.

Line 7 -14  – Please provide more detailed affiliation (Department, Chair etc.).

R: It was included.

Line 22, 27 – Please expand SSB, OR and CI -

R: Done.

Line 30 – I dont see purpose of using numbers in keywords section. R: We deleted these numbers.

Line 57 , 180, 250  – please use LAC abbreviation as implemented in abstract R: Corrected.

Line 180 – please verify whether data were obtained from 181 LAC or 197 LAC (line 17)?

R: It was corrected and changed in the abstract. The correct number is 181.

Line 73 – a brief description of methodology applied should follow the aim of the study.

R: We thank the reviewer for this suggestion and added briefly some information regarding the data sources analyzed in the study because we provided these information in details in the Methods section)

Line 113 – “km2” instead of “km2” R: Corrected.

Line 113 – “gas” or “gasoline” ? R: Gasoline. We had corrected in the text.

Line 138 – please provide details of SAS 9.4 (manufacturer, country of origin, license etc. ) R: We added the information required.

Round 2

Reviewer 1 Report

Articles can be accepted in the current form